# Impaired Metabolic Flexibility in the Osteoarthritis Process: A Study on Transmitochondrial Cybrids

**DOI:** 10.3390/cells9040809

**Published:** 2020-03-27

**Authors:** Andrea Dalmao-Fernández, Jenny Lund, Tamara Hermida-Gómez, María E Vazquez-Mosquera, Ignacio Rego-Pérez, Francisco J. Blanco, Mercedes Fernández-Moreno

**Affiliations:** 1Grupo de Investigación en Reumatología, Instituto de Investigación Biomédica de A Coruña (INIBIC), Agrupación estratégica CICA-INIBIC, Complexo Hospitalario Universitario de A Coruña (CHUAC), Sergas. Universidade da Coruña (UDC), 15006 A Coruña, Spain; andrea.dalmao.fernandez@sergas.es (A.D.-F.); Tamara.hermida.gomez@sergas.es (T.H.-G.); maria.eugenia.vazquez.mosquera@sergas.es (M.E.V.-M.); ignacio.rego.perez@sergas.es (I.R.-P.); 2Section for Pharmacology and Pharmaceutical Biosciences, Department of Pharmacy, University of Oslo, 0363 Oslo, Norway; jenny.lund@farmasi.uio.no; 3Centro de investigación biomédica en Red, Bioingeniería, Biomateriales y Nanomedicina (CIBER-BBN), 28029 Madrid, Spain

**Keywords:** osteoarthritis, transmitochondrial cybrids, energy metabolism, metabolic flexibility

## Abstract

Osteoarthritis (OA) is the most frequent joint disease; however, the etiopathogenesis is still unclear. Chondrocytes rely primarily on glycolysis to meet cellular energy demand, but studies implicate impaired mitochondrial function in OA pathogenesis. The relationship between mitochondrial dysfunction and OA has been established. The aim of the study was to examine the differences in glucose and Fatty Acids (FA) metabolism, especially with regards to metabolic flexibility, in cybrids from healthy (N) or OA donors. Glucose and FA metabolism were studied using D-[^14^C(U)]glucose and [1-^14^C]oleic acid, respectively. There were no differences in glucose metabolism among the cybrids. Osteoarthritis cybrids had lower acid-soluble metabolites, reflecting incomplete FA β-oxidation but higher incorporation of oleic acid into triacylglycerol. Co-incubation with glucose and oleic acid showed that N but not OA cybrids increased their glucose metabolism. When treating with the mitochondrial inhibitor etomoxir, N cybrids still maintained higher glucose oxidation. Furthermore, OA cybrids had higher oxidative stress response. Combined, this indicated that N cybrids had higher metabolic flexibility than OA cybrids. Healthy donors maintained the glycolytic phenotype, whereas OA donors showed a preference towards oleic acid metabolism. Interestingly, the results indicated that cybrids from OA patients had mitochondrial impairments and reduced metabolic flexibility compared to N cybrids.

## 1. Introduction

Osteoarthritis (OA) is the most frequent joint disease, but the etiopathogenesis of OA is not entirely understood. It is a heterogeneous disorder where genetics as well as biomechanical, endocrine, and inflammatory effects may be involved in its origin [1]. With acceptance of the joint as an organ, the pathogenesis of OA has been viewed as a complex process that involves cartilage degradation, synovial inflammation, subchondral sclerosis, muscular atrophy, and ligament damage; all culminating in joint dysfunction [2,3]. Recently, OA has been considered an illness with a low-grade chronic inflammation due to the synthesis and release of several cytokines and matrix metalloproteinases, leading to failure of the cartilage repair system and further exacerbation of the dysfunction [3,4].

The maintenance of energy homeostasis requires nutrient sensing, signaling, trafficking, storage, and oxidation, depending on substrate availability and energy demand [5]. Under stress, most cells adapt and are able to change their energy metabolism, either by increasing catabolic pathways when substrate availability is high or by increasing anabolic processes to respond to low nutrient intake [6]. The ability to respond and adapt to changes in metabolic demand has been described as metabolic flexibility [7,8]. This term is well described as a healthy characteristic in skeletal muscle [7,9]. Metabolic flexibility is associated with well-functioning mitochondria due to the fact of their role in the last steps of both glucose and FA metabolism [5]. Metabolic inflexibility implies mitochondrial dysfunction and, depending on substrate availability, impaired glucose or FA oxidation which further leads to nutrient accumulation [7,8].

During OA, it has been established a relationship between mitochondrial dysfunction and cellular damage due to the impairments in mitochondrial function and metabolic alterations, such as fragmented mitochondrial mass, increased production of reactive oxygen species (ROS) production, and apoptosis but lower ATP production, stress response mechanisms, and autophagy [10,11,12,13,14,15]. Several studies have suggested that free fatty acids (FFAs) may play an important role in OA development. In these studies, OA is associated with a high accumulation of lipids in cartilage [16,17]. Oleic, palmitic, and linoleic acid are the predominant FAs found in this tissue, accounting for approximately 85% of the total FA content [18]. Also, it has been established increased FA accumulation with progression of OA [18,19]. This ectopic FA accumulation likely leads to lipotoxicity and contributes to the cellular dysfunction [20]. Thus, the literature implies a relation between cartilage alterations and lipid metabolism [18,19].

Continent-specific mtDNA polymorphisms, known as mtDNA haplogroups, have been correlated by several OA population studies [21,22,23,24]. Some of these mtDNA polymorphisms modulate mitochondrial function that influence the behavior of the cell. Haplogroup J is biochemically different from haplogroup H, being the most efficient in terms of oxidative phosphorylation (OXPHOS) and lower ROS production. Therefore, it has been hypothesized that subjects with mtDNA haplogroup J confer protection in aging diseases such as OA [24,25]. Evidence suggests that mtDNA haplogroup J protects against knee and hip incidence and progression of OA [26]. Transmitochondrial cybrids are a useful in vitro model to study the effects of having varying mtDNA, as they have a uniform nuclear background [27]. They have been used in the study of a number of diseases where mitochondrial dysfunction is prominent, including Parkinson’s disease [28], Alzheimer’s disease [29], and Leber hereditary optic neuropathy [30]. Furthermore, several studies have successfully used cybrids to explore the role of mtDNA variation in disease pathogenesis by affecting processes such as mitochondrial proteostasis, ROS production, and mitochondrial dysfunction [31,32].

The purpose of the present work was to examine the glucose and FA metabolism, with particular focus on metabolic flexibility, in cybrids from healthy (N) or OA donors. Furthermore, the role of mtDNA haplogroups H and J was studied.

## 2. Materials and Methods

### 2.1. Subjects

Platelets from N and OA donors carrying mtDNA haplogroups H or J (N-H, N-J, OA-H, or OA-J) were obtained from samples belonging to the Sample Collection for the Research of Rheumatic Diseases, created by Blanco and registered in the Spanish National Biobank Registry, Section of Collections (Register Code C.0000424). Written informed consent was obtained from all subjects and approval was obtained from the local Ethics Committee of the Galician Health Administration (CEIC). All procedures were conducted according to the principles expressed in the Declaration of Helsinki. 

### 2.2. Transmitochondrial Cybrids Preparation

Cybrids derived from the 143B.TK^-^Rho-0 cell line were established as described previously [27]. Haplogroup genotyping was used to verify that the mtDNA of the cybrids line was the same as in the donor platelets. The mtDNA haplogroups were assessed using established methods [26].

Transmitochondrial cybrids were kept in Dulbecco´s modified Eagle’s medium with 5.5 mM (low) glucose (Gibco, Grand Island, NY, USA) (from here on described as DMEM-glu). For experiments, cybrids were cultured in DMEM-glu, DMEM no glucose (Gibco) supplemented with 100 µM oleic acid (Sigma–Aldrich, Merck KGaA, Darmstadt, Germany) (from here on described as DMEM-ole) or DMEM low glucose supplemented with 100 µM oleic acid (consequently described as DMEM-glu/ole). All media were further supplemented with 10% fetal bovine serum (FBS), penicillin (100 U/mL), and streptomycin (100 μg/mL) (Gibco). The cells were incubated in a humidified 5% CO_2_ atmosphere at 37 °C. At least two clones were obtained of each cybrid for each experiment.

### 2.3. Basal Glucose and Fatty Acid (FA) Metabolism

#### 2.3.1. Substrate Oxidation Assay and Measurement of Acid-Soluble Metabolites (ASM)

To analyze glucose and FA metabolism, cybrids were cultured with 1 × 10^4^ cells/well on 96 well CellBIND^®^ microplates (Corning Life Sciences, Schiphol-Rijk, Netherlands) in DMEM-glu or DMEM-ole for 48 h. Thereafter, D-[^14^C(U)]glucose (0.5 µCi/mL, 200 µM) (PerkinElmer NEN^®^, Boston, MA, USA) or [1-^14^C]oleic acid (0.5 µCi/mL, 100 µM) (PerkinElmer NEN^®^) were given during 4 h CO_2_ trapping as described previously [33]. In brief, a 96 well UniFilter^®^ microplate (PerkinElmer, Shelton, CT, USA), activated for the capture of CO_2_ by addition of 1 M NaOH, was mounted on top of the 96 well plate. After incubation, the [1-^14^C]oleic acid cell medium was transferred to a multi-well plate, sealed, and frozen at −20 °C for later analysis of acid-soluble metabolites (ASMs), before the cells were washed and harvested in 0.1 M NaOH. The ^14^CO_2_ trapped in the filter, produced from cellular respiration and cell-associated (CA) radioactivity from glucose or oleic acid uptake, was measured by the addition of scintillation fluid (Ultima Gold XR, PerkinElmer) and counted on a 2450 MicroBeta^2^ scintillation counter (PerkinElmer). Measurement of ASMs, which reflects incomplete FA β-oxidation and mainly consist of tricarboxylic acid cycle metabolites, was performed using a method modified from Skrede et al. [34]. From the [1-^14^C]oleic acid medium, 100 µL was transferred to an Eppendorf tube and precipitated with 300 µL cold HClO_4_ (1 M) and 30 µL BSA (6%). Thereafter, the tube was centrifuged at 10,000 rpm for 10 min at 4 °C before 200 µL of the supernatant was counted by liquid scintillation on Packard Tri-Carb 1900 TR (PerkinElmer). All results were adjusted for protein content, measured by the Bio-Rad protein assay using a VICTOR™ *X*4 Multilabel Plate Reader (PerkinElmer).

For glucose metabolism, the sum of ^14^CO_2_ and remaining CA radioactivity was taken as a measurement of total cellular uptake of substrate: CO_2_ + CA. Fractional glucose oxidation was calculated as CO_2_/(CO_2_ + CA). For calculation of total cellular oleic acid uptake, ASMs were also taken into account. The total uptake and fractional oxidation of oleic acid were thus calculated as CO_2_ + CA + ASM and CO_2_/(CO_2_ + CA + ASM), respectively. Fractional oxidation represents the proportion of uptake that goes to oxidation and may or may not correlate with complete cellular oxidation. 

#### 2.3.2. Scintillation Proximity Assay (SPA)

To measure glucose and FA accumulation, cells were cultured with 1 × 10^4^ cells/well on Cytostar-T^®^ 96 well scintillation microplate or 96-Scintiplate^®^ (PerkinElmer), respectively, in DMEM-glu or DMEM-ole for 24 h. Thereafter, accumulation of D-[^14^C(U)]glucose (0.5 µCi/mL, 200 µM) or [1-^14^C]oleic acid (0.5 μCi/mL, 100 µM) was measured by scintillation proximity assay (SPA) over 24 h as previously described [33]. In brief, cells were incubated during 24 h in the presence of radiolabeled substrates with measurements at 0, 2, 4, 6, 8, and 24 h on a 2450 MicroBeta^2^ scintillation counter (PerkinElmer). The radiolabeled substrates taken up will accumulate in the adherent cells and become concentrated close to the scintillator embedded in the plastic bottom of each well. After the 24 h accumulation, the cells were harvested in 0.1 M NaOH and protein content in the lysates were measured by the Bio-Rad protein assay using a VICTOR™ *X*4 Multilabel Plate Reader (PerkinElmer). 

### 2.4. Metabolic Flexibility

To study the competition between glucose and FA metabolism, two different protocols were established. In the first protocol, cells were cultured in DMEM-glu or DMEM-glu/ole, both containing 10% FBS, for 30 h. Thereafter, the media were changed to DMEM-glu/ole supplemented with 0.5% FBS the last 18 h before 4 h CO_2_ trapping with D-[^14^C(U)]glucose (0.5 µCi/mL, 200 µM). In the second protocol, cells were cultured as in the first protocol but during CO_2_ trapping, two inhibitors of glucose and FA metabolism, 20 µM UK5099 (Sigma–Aldrich) or 10 µM etomoxir (Sigma–Aldrich), respectively, were added. The UK5099 and etomoxir were dissolved in DMSO and water, respectively, to create concentrated stocks solutions. Thus, their respective controls were DMSO and water alone.

### 2.5. MTT Assay 

To assess if UK5099 or etomoxir affected cell viability, cells were cultured in the presence of UK5099 (2, 5, 10, 15, or 20 µM) or etomoxir (5, 10, 20, or 40 µM) for 4 h or 24 h. Thereafter, proliferation was determined using a CellTiter 96^®^ AQueous Non-Radioactive Cell Proliferation Assay (Promega, Madison, WI, USA) according to the manufacturer’s instructions. Absorbance was measured at 570 nm on a Tecan Infinite 200 microplate reader (Tecan, Männedorf, Switzerland). The analysis showed that these compounds, at 20 and 10 µM, respectively, over 4 h did not affect cell viability (data not shown). 

### 2.6. Detection of Anion Superoxide Production (O_2_^−^)

Cells were cultured with 9 × 10^4^ cells/well on Corning^®^ CellBIND^®^ 6 well plates (Corning) in DMEM-glu/ole for 48 h. The last 4 h, some parallels were incubated in the presence of UK5099 (20 µM) or etomoxir (10 µM). To measure mitochondrial anion superoxide (O_2_^−^) production, cells were treated during 30 min with 5 µM MitoSOX™ Red (Invitrogen, Carlsbad, CA, USA). MitoSOX™ Red reagent permeates living cells and selectively targets the mitochondria. It is rapidly oxidized by O_2_^−^ but not by other reactive oxygen species (ROS). Cells were harvested by trypsin release and resuspended in saline solution prior to analysis on a FACsCalibur flow cytometer (Becton Dickinson, New Jersey, USA). Per assay 1 × 10^4^ cells were measured and data were analyzed using the CellQuest software (Becton Dickinson). 

### 2.7. Evaluation of Lipid Droplet (LD) Formation

#### 2.7.1. Lipid Distribution by Thin Layer Chromatography (TLC)

Cells were cultured with 1.5 × 10^5^ cells/well on Corning^®^ CellBIND^®^ 12 well plates (Corning) in DMEM with 25 mM glucose for 24 h before being incubated with [1-^14^C]oleic acid (0.5 µCi/mL, 100 µM) for another 24 h. Thereafter, the cells were washed twice with PBS and harvested in 0.1% SDS. Cellular lipid distribution was analyzed as previously described [35] by extraction of the homogenized cell fraction, separation of lipids by thin layer chromatography (TLC) using a non-polar solvent mixture of hexane:diethyl ether:acetic acid (65:35:1), and quantification by liquid scintillation on Tri-Carb 1900 TR (PerkinElmer). Total protein was measured by Pierce™ BCA Protein Assay Kit (Thermo Fisher Scientific, Madrid, Spain) using a VICTOR™ *X*4 Multilabel Plate Reader (PerkinElmer).

#### 2.7.2. Staining Neutral Lipids by Immunofluorescence

Cells were cultured with 3 × 10^4^ cells/well on 4 well chamber slides (Millipore, Billerica, MA, USA) for 48 h in DMEM with 25 mM glucose or DMEM-ole. Thereafter, the cells were fixed in 4% paraformaldehyde (Sigma–Aldrich) for 10 min at room temperature, followed by incubation in 0.1% Tween20 (Sigma–Aldrich) for 5 min. The LD staining was performed with a 1:10,000 dilution of a neutral lipid dye LD540 (kindly provided by Dr. Thiele), for 30 min [36]. This step was followed by 5 min incubation with the nuclear dye Hoechst 33,258 (Sigma–Aldrich). After washing with PBS, cover slips were mounted on microscopy chamber slides using Prolong Gold Antifade Reagent Mountant (Thermo Fisher Scientific). Fluorescence was visualized and photographed under a Nikon AR-1 confocal microscope (Nikon, Tokyo, Japan).

### 2.8. Flow Cytometry

The intensity of LD540 fluorescent was also evaluated on a FACsCalibur flow cytometer (Becton Dickinson). Cells were cultured with 9 × 10^4^ cells/well on Corning^®^ CellBIND^®^ 6 well plates (Corning) and treated as described above. Cells were harvested by trypsin release and resuspended in saline solution prior to analysis by flow cytometry. 1 × 10^4^ cells per assay were measured. Data were analyzed using CellQuest software (Becton Dickinson). Results were expressed as median of fluorescence (AU) from three independent experiments.

### 2.9. Quantitative Real-Time PCR (RT-PCR)

The RNA extraction was achieved using TRIzol (Life Technologies) according to the manufacturers protocol. 0.5 µg of RNA was reversely transcribed into cDNA using an NZY First-Strand cDNA Synthesis Kit (NZYTech, Lisbon, Portugal) following the manufacturer´s instructions. The RT-PCR was performed in a LightCycler 480-II Instrument (Roche, Mannheim, Germany) using TaqMan Universal Master Mix (Roche). Analysis of the results was carried out using Qbase+ version 2.5 software (Biogazelle, Ghent, Belgium). Gene expression was calculated relative to the housekeeping gene ribosomal protein L13A (*RPL13A*). Sequence primers, probe, and PCR conditions are available upon request.

### 2.10. Statistical Analyses

Data are presented as mean ± standard error mean (SEM) from three independent experiments with a minimum of three observations unless stated otherwise. Statistical analyses were performed using GraphPad Prism software version 6.01 (GraphPad, La Jolla, CA, USA). The unpaired *t-*test was used to evaluate differences between N and OA groups, whereas the Mann–Whitney test was used to evaluate differences between haplogroups H and J. The paired *t-*test was used to evaluate the effect of different treatments compared to basal. Differences with *p*-values ≤ 0.05 were considered to be statistically significant.

## 3. Results

### 3.1. Basal Glucose and FA Metabolism

#### 3.1.1. Basal Glucose and FA Metabolism

Basal glucose and oleic acid metabolism were analyzed in cybrids by measuring acute (4 h) oxidation and uptake as well as time-course accumulation over 24 h (Figure. 1). There were no changes in basal glucose metabolism between N and OA cybrids (Figure 1A,B). Furthermore, when evaluating the differences between the two haplogroups, OA-J had higher cell-associated glucose than OA-H (Figure 1C); there was a non-significant tendency towards higher complete oxidation in OA-J compared to OA-H cybrids (*p* = 0.0581 Figure 1C).

Examination of basal oleic acid metabolism (Figure 1E–H) showed that OA cybrids had lower ASM, reflecting incomplete FA β-oxidation, compared to N cybrids (Figure 1E), whereas the other parameters were unchanged (Figure 1E,F). When evaluating the role of haplogroups, a lower complete and fractional oleic acid oxidation was observed in N-J compared to N-H cybrids (Figure 1G). Furthermore, oleic acid accumulation was overall higher in N-J than N-H cybrids the first 8 h of the time-course substrate incorporation (Figure 1H). 

#### 3.1.2. Comparison between Basal Glucose and FA Metabolism 

In order to see differences in degree of glucose compared to oleic acid, we performed a comparative analysis between the synonymous data obtained from the substrate oxidation assay with the two substrates as reported separately above. We observed that both N and OA cybrids had higher CA, but lower complete and fractional oxidation of oleic acid compared to glucose, indicating a preference towards glucose oxidation. This was reflected within the N cybrids, where both haplogroups had lower complete and fractional oxidation but higher CA of oleic acid. However, within the haplogroups of OA cybrids, the OA-J had both lower complete and fractional oxidation of oleic acid, whereas OA-H only had lower fractional oxidation (Table 1).

### 3.2. Metabolic Flexibility

The results obtained in basal experiments showed that cybrids from N and OA donors oxidize more glucose, whereas they have higher CA of oleic acid. Thus, we wanted to examine how addition of oleic acid affected the cybrids´ glucose metabolism. Compared to basal (DMEM-glu), oleic acid increased complete oxidation and total cellular uptake of glucose in N but not in OA cybrids (Figure 2A). Complete and fractional glucose oxidation were lower in OA cybrids compared to N (Figure 2A). Evaluating the results from the functional studies, gene expression analyses were performed. mRNA expression of carnitine palmitoyltransferase 1B (*CPT1B*) and mitochondrial pyruvate carrier 2 (*MPC2*) were lower in OA than in N cybrids (Figure 2B). The mitochondrial superoxide (O_2_^−^) production was higher in OA cybrids when oleic acid was present in the culture medium (Figure 2C). Combined, these results indicate that N cybrids are more flexible than OA when the supply of energy changes.

To further evaluate the metabolic flexibility, cells were cultured in DMEM-glu/ole and glucose and oleic acid metabolism in the presence or absence of the mitochondrial inhibitors etomoxir (Figure 3A) or UK5099 (Figure 3B), respectively. Etomoxir inhibits FA oxidation by targeting *CPT1B,* a mitochondrial carrier of FAs, whereas UK5099 is an inhibitor of *MPC* [37,38,39]. Inhibition by etomoxir increased complete oxidation, CA, and total cellular uptake of glucose in N cybrids compared to basal, but not in OA cybrids where fractional glucose oxidation was decreased (Figure 3A). Inhibition by UK5099 increased complete and fractional oleic acid oxidation in OA cybrids compared to basal but not in N cybrids (Figure 3B). Furthermore, N cybrids had a slight tendency towards increased glucose oxidation (ns, *p* = 0.0675) compared to OA cybrids (Figure 3A), whereas these cells oxidized more oleic acid compared to N cybrids when cultured in the presence of UK5099 (Figure 3B). Additionally, cells cultured in presence of UK5099 had higher O_2_^−^ production in N and OA cybrids, whereas gene expression of mitofusin 2 (*MFN2*) was increased in N but not OA cybrids (Figure 3C,D). No effect of etomoxir was observed on the levels of O_2_^−^ production and *MFN2* mRNA expression (Figure 3C,D). Mitochondrial fusion plays a critical role in maintaining functional mitochondria when cells experience a metabolic change. These results suggest that N cybrids are able to increase glucose metabolism when the FA pathway is blocked, whereas OA cybrids were only able to increase oleic acid oxidation when the glucose pathway was blocked. Combined, this provides further evidence for higher metabolic flexibility in cybrids carrying mitochondria from healthy individuals compared to patients with OA.

### 3.3. Evaluation of Lipid Distribution and LD Formation 

To study the incorporation of oleic acid into lipids, TLC was performed. Most of the oleic acid was incorporated into triacylglycerol (TAG) and phospholipids (PL), accounting for approximately 70% and 25% of the total lipid incorporation, respectively (Figure 4A). The cybrids presented different lipid distribution patterns: N cybrids had higher incorporation into PL and cholesteryl ester (CE), but lower incorporation into TAG, compared to OA cybrids (Figure 4A). With TAG being the major contributor to total FA incorporation and LD formation, the differences among haplogroups were examined for this lipid class; OA-J showed higher TAG incorporation than OA-H cybrids (Figure 4B). 

Taking into account that TAG is mainly stored as LDs, intracellular storage depots of neutral lipids [40], LD formation was also analyzed. Addition of oleic acid to cell culture medium was used as an efficient strategy to enhance and study the intracellular accumulation of LDs. Compared to basal, LD formation increased in both groups (Figure 4C,D). Furthermore, N cybrids incorporated less oleic acid into LDs than OA (Figure 4D). In coherence with higher TAG incorporation in OA-J cybrids, LD formation was higher in OA-J compared to OA-H and N-J haplogroups (Figure 4E).

## 4. Discussion

The results obtained in this study enhance our understanding of mitochondrial metabolism in OA and suggests a mitochondrial dysfunction related to impaired metabolic flexibility during the OA process.

To study if different substrates could influence the metabolism in N and OA cybrids, basal metabolism of glucose and oleic acid were examined. No differences in glucose metabolism were observed between N and OA cybrids. However, examination of basal oleic acid metabolism showed that OA cybrids had lower ASM, reflecting incomplete FA metabolism compared to N cybrids. Although glucose is the main energy source in chondrocytes [41], lipids are also necessary, both to be utilized for energy and to be incorporated as structural components and signaling molecules [42]. Thus, lower ASM levels indicate less efficient mitochondria from OA donors, supporting the data obtained in chondrocytes where it has been established a relationship between mitochondrial dysfunction and OA processes [10,43]. 

When the role of mtDNA haplogroups in basal metabolism was analyzed, only OA-J cybrids showed higher cell-associated glucose, whereas N-J cybrids showed lower FA oxidation compared to their respective H haplogroup. This could be a result of the characteristic bioenergetics of mtDNA haplogroup J. In agreement with data from a previous study, cybrids from healthy donors carrying haplogroup J seemed to be able to increase cell survival by mitochondrial respiration and glycolysis, decreasing ATP and reactive oxygen species production [24]. 

Glucose and FA metabolism are two of the most significant mechanisms in order to obtain energy [5,44]. The cybrids were mainly glycolytic, favoring glucose over oleic acid for oxidation. Despite this, OA cybrids had lower mRNA levels of *CPT1*, a mitochondrial transporter of FAs. Others have reported that chondrocytes also mainly are glycolytic, with 25% of the ATP produced through OXPHOS [45]. When co-culturing the cells with glucose and oleic acid (with or without etomoxir or UK5099), the results showed that only N cybrids increased glucose metabolism, thus remaining glycolytic even in the presence of oleic acid and etomoxir. The OA cybrids were, on the other hand, unable to adapt their glucose metabolism with the changed energetic conditions, thus suggesting a mitochondrial impairment and preference of FAs for oxidation. The OA development has been related to mitochondrial dysfunction and cellular damage due to the impairments in mitochondrial function [10,11,12,13,14,15]. Superoxide production was found to be higher in OA than N cybrids. Mitochondrial ROS production is directly dependent on mitochondrial function and an increase of this parameter has been linked to the OA pathogenesis [46]. Furthermore, it has also been suggested that mitochondrial dysfunction induces FA oxidation and oxidative stress [47,48,49]. Thus, this study may confirm the role of mitochondrial dysfunction in OA as seen by the results from OA cybrids with preference of FAs for oxidation and increased ROS production. 

Analyzing the effect of inhibitors in the cybrids, UK5099 increased ROS levels in both N and OA cybrids, but only N cybrids had higher mRNA expression of *MFN2*. Mitochondrial dynamics are altered by different metabolic status in the cell and this is controlled by fission and fusion processes [50]. In addition, mitochondrial fusion, through the phosphorylation of *MFN2*, controls the autophagy of dysfunctional mitochondria to maintain a healthy status [51]. Thus, these results are in line with the fact that OA cybrids showed lower adaptative response under metabolic stress compared to N cybrids. 

A shortcoming of the study is the limited examination of cell viability and functionality, which requires future studies. Furthermore, it will be relevant to evaluate the mitochondrial functionality and integrity more extensively.

Taken together, the data demonstrate that cybrids from OA donors behave differently from N donors. Particularly interesting was the finding that OA cybrids showed lower metabolic flexibility and increased oxidative stress compared to N cybrids when exposed to changed nutrient supply. This concept was described in bovine chondrocytes in 2015, where N chondrocytes showed the healthy characteristic of metabolic flexibility, associated with cell survival during nutrient stress by upregulation of mitochondrial respiration and reduction in ROS production [14]. Although great progress has been made in understanding OA, the current knowledge of metabolic flexibility in chondrocytes is still at an early stage. This study complements the exiting knowledge, and combined it suggests that poor regulation of metabolic flexibility could be linked to the etiology of OA. 

Finally, in the present study it was observed that the transmitochondrial cybrids preferred to accumulate oleic acid compared to glucose. When lipid distribution was studied, a lower incorporation of oleic acid into PLs and CE was observed in OA compared to N cybrids. However, interestingly, more oleic acid was incorporated into TAG in OA cybrids, and they also showed higher LDs formation compared to N cybrids. Others have reported that free FAs can be incorporated by cartilage and chondrocytes into phosphatidylcholine, phosphatidylethanolamine, phosphatidylinositol, and TAG [16,17,18,19,20]. The PLs are important for normal cartilage maintenance and TAG for storage in lipid droplets (LDs) [42]. In the second case, it has been hypothesized that FAs stored in LDs may serve a protective role against the stressors [52,53]. Surprisingly, haplogroup J was the major contributor to this TAG accumulation and showed higher formation of LDs. This finding supports the hypothesis that subjects with mtDNA haplogroup J confer protection in OA, where altered lipid metabolism may be a risk factor [16,24,25]. As mentioned before, different studies suggested that mtDNA haplogroup J protects against incidence and progression of OA in knees and hips [26]. At a molecular level, the relationship between haplogroup J and protection of OA remains to be fully elucidated. However, it is reasonable to think that N-J and OA-J accumulate more LDs than N-H and OA-H in order to protect the cell against impairments in FA metabolism as a protective mechanism against incidence and progression of OA.

## 5. Conclusions

The transmitochondrial cybrids preferred to oxidize glucose and accumulate lipids. Comparison of N and OA cybrids showed no differences in basal glucose metabolism but lower incomplete FA β-oxidation in OA cybrids associated with lower gene expression of *CPT1B*. Furthermore, OA cybrids accumulated more TAG and LDs than N cybrids. Importantly, an impaired metabolic flexibility was observed in OA cybrids, providing further evidence towards a mitochondrial dysfunction in the OA pathogenesis.

## Figures and Tables

**Figure 1 cells-09-00809-f001:**
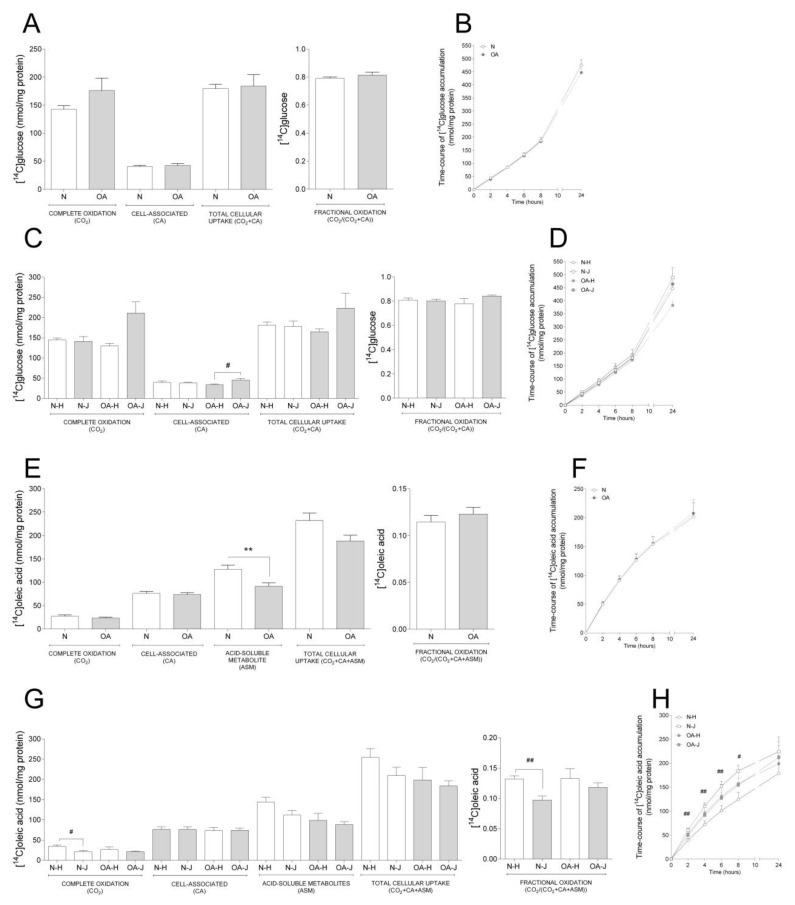
Basal glucose and fatty acid metabolism. Cybrids were cultured for 48 h in DMEM-glu (A–D, i.e., DMEM 5.5 mM glucose) or DMEM-ole (E–H, i.e., DMEM no glucose supplemented with 100 µM oleic acid). Thereafter, basal glucose and oleic acid metabolism were evaluated using D-[^14^C(U]glucose (0.5 µCi/mL, 200 µM) or [1-^14^C]oleic acid (0.5 µCi/mL, 100 µM), respectively, and 4 h substrate oxidation assay (A, C, E, and G) or 24 h SPA (B, D, F, and H). (**A**,**B**) Basal glucose metabolism in healthy (N) and osteoarthritic (OA) cybrids. (**C**,**D**) Basal glucose metabolism in cybrids carrying haplogroups H or J. (**E**,**F**) Basal oleic acid metabolism in N and OA cybrids. (**G**,**H**) Basal oleic acid metabolism in cybrids carrying haplogroups H or J. N and OA data included the values for haplogroups H and J combined. All data were obtained from three independent experiments, each with four replicates and two clones per donor. Values are presented as mean ± SEM in nmol/mg protein. * Statistically significant difference between N and OA cybrids (** *p* ≤ 0.005, unpaired *t*-test). ^#^ Statistically significant difference between haplogroups (^#^
*p* ≤ 0.05, ^##^
*p* ≤ 0.005, Mann–Whitney test).

**Figure 2 cells-09-00809-f002:**
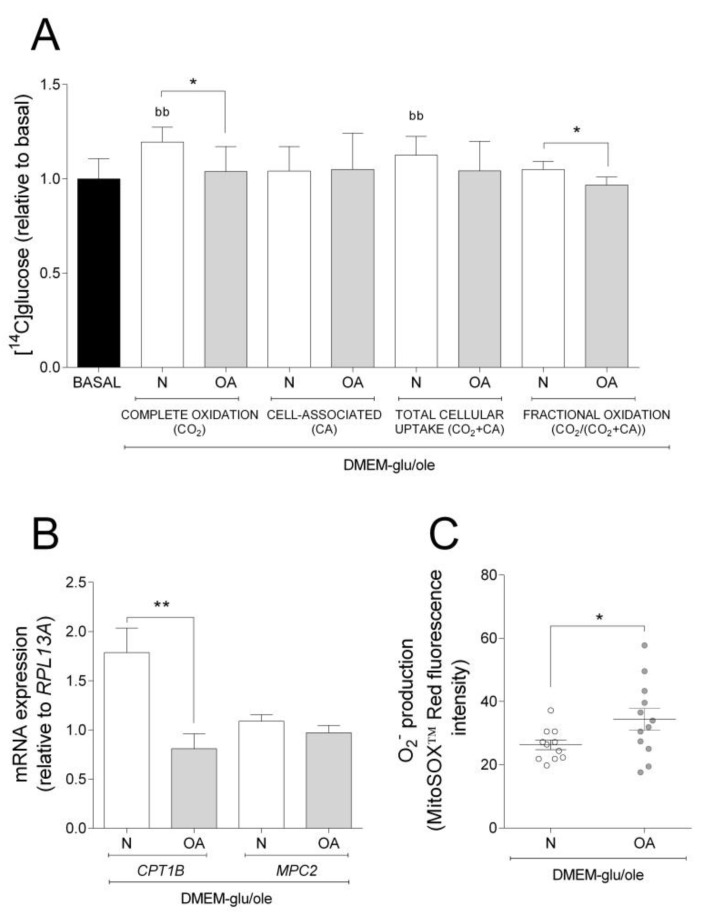
Effect of oleic acid on glucose metabolism, mRNA expression, and O_2_^−^ production. Effect of culturing cells in DMEM-glu/ole (i.e., DMEM 5.5 mM glucose supplemented with 100 µM oleic acid) compared to DMEM-glu (basal, i.e., DMEM 5.5 mM glucose). (**A**) Glucose metabolism. (**B**) mRNA expressions of carnitine palmitoyltransferase 1B (*CPT1B*) and mitochondrial pyruvate carrier 2 (*MPC2*) normalized to the expression of *RPL13A*. (**C**) Anion superoxide production (O_2_^−^). N and OA data included the values for haplogroups H and J combined. All data were obtained from three independent experiments, each with four replicates and two clones per donor. Values are presented as mean ± SEM relative to basal. * Statistically significant difference between N and OA cybrids (* *p* ≤ 0.05, ** *p* ≤ 0.01, unpaired *t*-test). ^b^ Statistically significant versus basal (^bb^
*p* ≤0.01, paired *t*-test). N, healthy cybrids; OA, osteoarthritis cybrids.

**Figure 3 cells-09-00809-f003:**
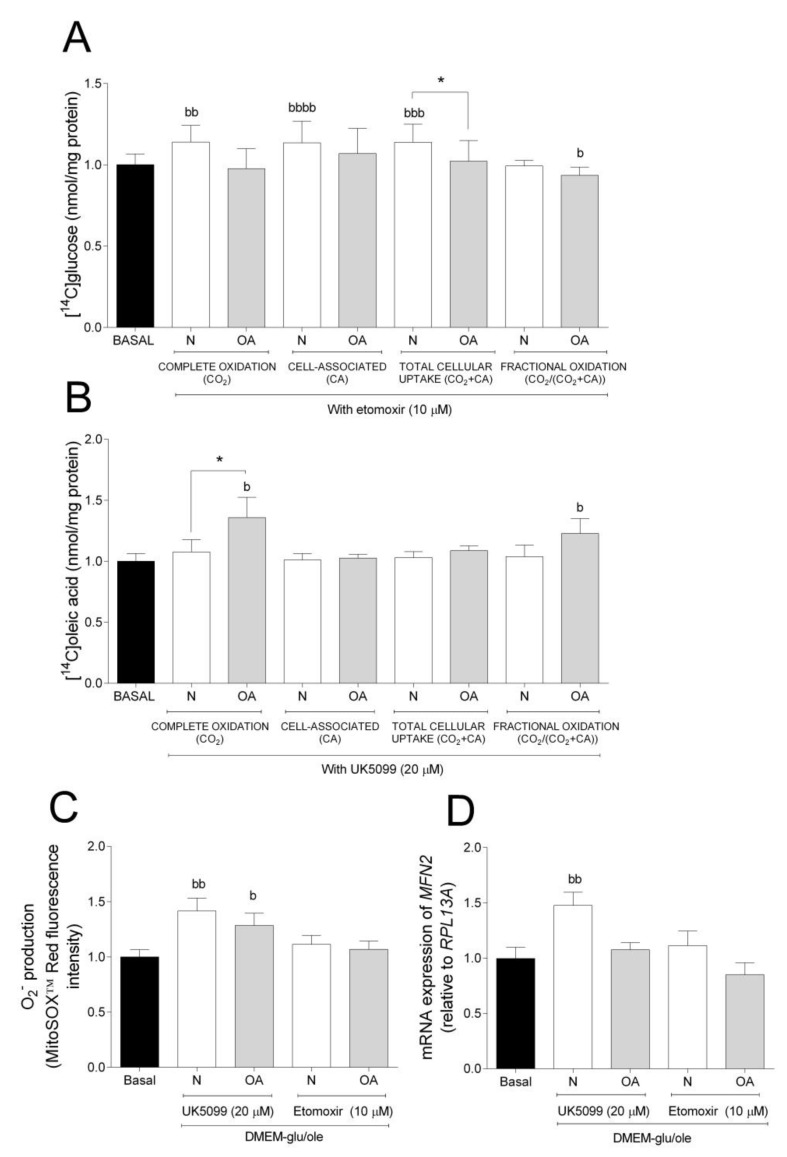
Metabolic flexibility. (**A**) Effect of 10 µM etomoxir on glucose metabolism. (**B**) Effect of 20 µM UK5099 on oleic acid metabolism. (**C**) Effect of 10 µM etomoxir or 20 µM UK5099 on anion superoxide (O_2_^−^) production. (**D**) Effect of 10 µM etomoxir or 20 µM UK5099 on mRNA expression of mitofusin 2 (*MFN2* N and OA data included the values for haplogroups H and J combined. All data were obtained from three independent experiments, each with four replicates and two clones per donor. Values are presented as mean ± SEM relative to basal (DMEM-glu/ole, i.e., DMEM 5.5 mM glucose supplemented with 100 µM oleic acid). * Statistically significant difference between N and OA cybrids (* *p* ≤ 0.05, unpaired *t*-test). ^b^ Statistically significant versus basal (^b^
*p* ≤ 0.05, ^bb^
*p* ≤ 0.01, ^bbb^
*p* ≤ 0.001, ^bbbb^
*p* ≤ 0.0001 paired *t*-test). N, healthy cybrids; OA, osteoarthritis cybrids.

**Figure 4 cells-09-00809-f004:**
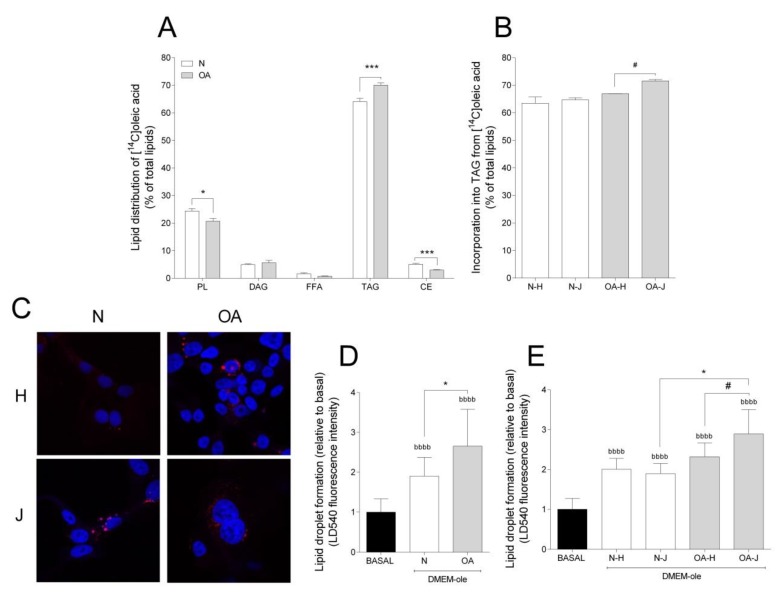
Lipid distribution and lipid droplet formation. (**A**) Lipid distribution in healthy (N) and osteoarthritis (OA) cybrids after 24 h incubation with [1-^14^C]oleic acid. PL, phospholipid; DAG, diacylglycerol; FFAs, free fatty acids; TAG, triacylglycerol; CE, cholesteryl ester. (**B**) Incorporation of [1-^14^C]oleic acid into TAG in N and OA cybrids carrying haplogroup H or J. (**C**) One representative image of cybrids after culturing in DMEM-ole (i.e., DMEM no glucose supplemented with 100 mM oleic acid) for 48 h before staining lipid droplets with LD540 (red) and nuclei with Hoechst 33,258 (blue). (**D**,**E**) Quantification of lipid droplet formation by flow cytometry in N and OA cybrids (**D**), carrying haplogroups H or J (**E**) relative to basal (DMEM 25 mM glucose). N and OA data included the values for haplogroups H and J combined. All data were obtained from three independent experiments, each with at least three replicates and two clones per donor. Values are presented as mean ± SEM relative to total lipids (A and B) or relative to basal (D and E). * Statistically significant difference between N and OA cybrids (* *p* ≤ 0.05, *** *p* ≤ 0.001, unpaired *t*-test). ^#^ Statistically significant difference between haplogroups (^#^
*p* ≤ 0.05, Mann–Whitney test). ^b^ Statistically significant versus basal (^bbbb^
*p* ≤ 0.0001, paired *t*-test).

**Table 1 cells-09-00809-t001:** Comparison between basal glucose and FA metabolism from healthy (N) and osteoarthritis (OA) cybrids and carrying haplogroups H or J.

	Complete Subtrate Oxidation (CO_2_) (nmol/mg Protein)		Cell-Associated (CA) (nmol/mg Protein)		Total Cellular Uptake (CO_2_ + CA) (nmol/mg Protein)		Fractional Subtrate Oxidation(CO_2_/(CO_2_ + CA))	
	[^14^C]glucose	[^14^C]oleic acid	[^14^C]glucose	[^14^C]oleic acid	[^14^C]glucose	[^14^C]oleic acid	[^14^C]glucose	[^14^C]oleic acid
**N**	58.52 ± 3.86	**27.44 ± 3.15 ^aaaa^**	45.92 ± 5.73	**76.29 ± 4.46 ^aaaa^**	104.44 ± 9.29	104.00 ± 7.07	0.57 ± 0.02	**0.25 ± 0.02 ^aaaa^**
**OA**	66.28 ± 8.47	**23.31 ± 2.29 ^aaa^**	50.41 ± 9.19	**73.76 ± 3.93 ^a^**	116.60 ± 17.42	97.08 ± 5.56	0.59 ± 0.02	**0.23 ± 0.01 ^aaaa^**
**N-H**	60.10 ± 3.93	**34.01 ± 3.83 ^aaa^**	49.69 ± 7.40	**76.30 ± 6.89 ^a^**	109.78 ± 10.86	110.40 ± 10.56	0.56 ± 0.02	**0.30 ± 0.01 ^aaaa^**
**N-J**	56.95 ± 7.00	**20.87 ± 3.41 ^aaa^**	42.16 ± 9.17	**76.27 ± 6.33 ^a^**	99.11 ± 15.8	97.59 ± 9.61	0.60 ± 0.04	**0.20 ± 0.01 ^aaaa^**
**OA-H**	56.48 ± 11.78	26.84 ± 6.38	46.73 ± 17.53	73.12 ± 8.23	102.80 ± 28.94	99.96 ± 13.94	0.57 ± 0.06	**0.26 ± 0.01^a^**
**OA-J**	71.18 ± 11.42	**21.55 ± 1.61 ^aa^**	52.26 ± 11.80	74.08 ± 4.84	123.4 ± 23.07	95.63 ± 5.85	0.59 ± 0.02	**0.22 ± 0.01 ^aaaa^**

Values are presented as mean ± SEM. A statistically significant versus [^14^C]glucose (^a^
*p* ≤ 0.05, ^aa^
*p* ≤ 0.005, ^aaa^
*p* ≤ 0.001, ^aaaa^
*p* ≤ 0.0001, unpaired *t*-test). N, healthy; OA, osteoarthritis.

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
