# Peer review of "Impaired Metabolic Flexibility in the Osteoarthritis Process: A Study on Transmitochondrial Cybrids"

_cells, 2020, doi:10.3390/cells9040809_

Round 1
Reviewer 1 Report
This manuscript describes detailed and thorough studies to examine how central metabolism is affected by mitochondria from normal and osteoarthritic (OA) donors. Toward this, the authors created transmitochondrial cybrids using established methodology. Cybrids came from normal and OA donors, as well as mitochondrial haplotypes H and J. The authors then measured basal glucose and fatty acid metabolism and accumulation using established biochemical methodology. Further studies examined viability, superoxide production, lipid droplit formation, and lipid distribution. The results show that there are differences in central metabolism between cybrids with normal and OA mitochondria, and that OA cybrids have lower metabolic flexibility and mitochondrial impairments. These data are likely important to understanding the biology of OA.
The only major concern that I have is that readers need to know if the data in Figures 1-3 come from cybrids of haplogroup H or J, or from mixed samples.
Minor Concerns include:
- L44: the word “the” can be removed before energy homeostasis.
- L126: the denominator of the equation should be listed in parentheses
- L158: a description of the mechanism by which MitoSOX detects superoxide should be added.
- Table 1—are there relevant units for these numbers (e.g. nanomoles/minute)?
Author Response
Response to Reviewer 1
This manuscript describes detailed and thorough studies to examine how central metabolism is affected by mitochondria from normal and osteoarthritic (OA) donors. Toward this, the authors created transmitochondrial cybrids using established methodology. Cybrids came from normal and OA donors, as well as mitochondrial haplotypes H and J. The authors then measured basal glucose and fatty acid metabolism and accumulation using established biochemical methodology. Further studies examined viability, superoxide production, lipid droplet formation, and lipid distribution. The results show that there are differences in central metabolism between cybrids with normal and OA mitochondria, and that OA cybrids have lower metabolic flexibility and mitochondrial impairments. These data are likely important to understanding the biology of OA.
The only major concern that I have is that readers need to know if the data in Figures 1-3 come from cybrids of haplogroup H or J, or from mixed samples.
Answer: Yes, the N and OA data in figures 1-4 show the results of haplogroup H and J combined.
Action: We have included the following sentence in legends to figures 1-4: “N and OA data included values for haplogroups H and J combined”.
Minor Concerns include:
- L44: the word “the” can be removed before energy homeostasis.
- Action: We followed the reviewer´s suggestion and have removed the word (p. 2, L44).
- L126: the denominator of the equation should be listed in parentheses.
- Action: We included the parentheses in the denominator of the suggested equation (p. 5, 127). This will also apply to the other similar equation, which also has been corrected accordingly (p. 5, L125). Table 1 (p. 11) and figures 1-3 (pp. 10, 12 and 14) have also been edited accordingly.
- L158: a description of the mechanism by which MitoSOX detects superoxide should be added.
- Answer: MitoSOX™ Red reagent permeates living cells and selectively targets the mitochondria. It is rapidly oxidized by superoxide but not by other reactive oxygen species.
- Action: We added the MitoSoxTM mechanism as explained above (p. 6, L164-166).
- Table 1—are there relevant units for these numbers (e.g. nanomoles/minute)?
- Answer: Yes, the units of the parameters described in the table were included in the table legend. Complete substrate oxidation (CO2), cell-associated (CA) and total cellular uptake (CO2+CA) are presented as mean ± SEM in nmol/mg protein. Fractional substrate oxidation (CO2/(CO2+CA)) does naturally not have a unit as it is a fraction.
- Action: We have included the parameters units in the table to facilitate the interpretation data (p. 11).
Reviewer 2 Report
Manuscript entitled „Impaired metabolic flexibility in the osteoarthritis process – a study on transmitochondrial cybrids” presents results from interesting study which focused on the association between mitochondrial dysfunction and osteoarthritis (OA). Specifically, the authors assessed the difference between glucose and fatty acids metabolism in mitochondrial cybrids from healthy and osteoarthritis donors, including also the impact of mitochondrial haplogroups. For this puropose, complex metabolic analyses were conducted using mitochondrial cybrids which were constructed using 143B.TH-Rho-0 cells (which do not have mt DNA) and platelets from donors (cells without nucleus) carrying two mtDNA haplogroups (H or J). The obtained data point to the reduced metabolic flexibility in OA related to fatty acids metabolism.
The manuscript is very well written, all sections are clear and concise. In general, I do not have any serious comments. I find these results novel and important in the field. I recommend the paper to be published in Cells.
Minor comments:
- Methods – please indicate the country for Biobank.
- Figures – axis titles could be bigger.
Author Response
Response to Reviewer 2
Manuscript entitled “Impaired metabolic flexibility in the osteoarthritis process – a study on transmitochondrial cybrids” presents results from interesting study which focused on the association between mitochondrial dysfunction and osteoarthritis (OA). Specifically, the authors assessed the difference between glucose and fatty acids metabolism in mitochondrial cybrids from healthy and osteoarthritis donors, including also the impact of mitochondrial haplogroups. For this puropose, complex metabolic analyses were conducted using mitochondrial cybrids which were constructed using 143B.TH-Rho-0 cells (which do not have mt DNA) and platelets from donors (cells without nucleus) carrying two mtDNA haplogroups (H or J). The obtained data point to the reduced metabolic flexibility in OA related to fatty acids metabolism.
Comments: The manuscript is very well written, all sections are clear and concise. In general, I do not have any serious comments. I find these results novel and important in the field. I recommend the paper to be published in Cells.
Answer: Thank you very much for your kind comments.
Minor comments:
- Methods – please indicate the country for Biobank.
- Action: Spain was country for the BioBank, this has now been stated in the manuscript (p. 3, L85).
- Figures – axis titles could be bigger.
- Action: We have increased the font size of axis titles in all figures from 12 to 16.
Reviewer 3 Report
The present work entitled: Impaired metabolic flexibility in the osteoarthritis process – a study on transmitochondrial cybrids, accurately describes the metabolism of glucose and fatty acids under normal conditions and in osteoarthritis.
The evaluation of cell viability and cell functionality is poorly evaluated. In addition to MTT, it would be useful to evaluate, possible cell death, Annexin V or TUNEL reaction, to assess if a part of the cells undergoes programmed cell death. Furthermore, it would be appropriate to evaluate the functionality and integrity of mitochondria under a confocal microscope with Acridine Orange (AO) and PI Nuclei Staining and NAO.
Author Response
Response to Reviewer 3
The present work entitled: Impaired metabolic flexibility in the osteoarthritis process – a study on transmitochondrial cybrids, accurately describes the metabolism of glucose and fatty acids under normal conditions and in osteoarthritis.
Comments: The evaluation of cell viability and cell functionality is poorly evaluated. In addition to MTT, it would be useful to evaluate, possible cell death, Annexin V or TUNEL reaction, to assess if a part of the cells undergoes programmed cell death. Furthermore, it would be appropriate to evaluate the functionality and integrity of mitochondria under a confocal microscope with Acridine Orange (AO) and PI Nuclei Staining and NAO.
Answer: Thank you very much for your interesting comment. We used the MTT assay to assess the effect of UK5099 and etomoxir on cell viability. The objective of the manuscript was to examine glucose and FA metabolism, with particular focus on metabolic flexibility, in cybrids from N and OA donors. Furthermore, we wanted to study the role of mtDNA haplogroups H and J. Comprehensive analysis of cell viability as well as mitochondrial morphology, functionality and cell death was not an objective of the present work. Although interesting this is not feasible to do within the allotted time and furthermore with the quarantine restrictions due to the current COVID-19 pandemic. We will keep your great suggestions in mind for future studies with these cells and have included a section in the discussion of the manuscript stating this as a limitation to the current study.
Action: We have commented on the lack of these experiments as limitations in the discussion of the revised manuscript (p. 18, L378-380).
Round 2
Reviewer 3 Report
I believe that the work can be published in the revised form